# Proteomic Comparison of Acute Myeloid Leukemia Cells and Normal CD34^+^ Bone Marrow Cells: Studies of Leukemia Cell Differentiation and Regulation of Iron Metabolism/Ferroptosis

**DOI:** 10.3390/proteomes13010011

**Published:** 2025-02-17

**Authors:** Frode Selheim, Elise Aasebø, Håkon Reikvam, Øystein Bruserud, Maria Hernandez-Valladares

**Affiliations:** 1Proteomics Unit of University of Bergen (PROBE), University of Bergen, Jonas Lies vei 91, 5009 Bergen, Norway; frode.selheim@uib.no (F.S.); mariahv@ugr.es (M.H.-V.); 2Acute Leukemia Research Group, Department of Clinical Science, University of Bergen, Jonas Lies vei 91, 5009 Bergen, Norway; elise.aasebo@helse-bergen.no (E.A.); hakon.reikvam@uib.no (H.R.); 3Section for Hematology, Department of Medicine, Haukeland University Hospital, 5009 Bergen, Norway; 4Department of Physical Chemistry, University of Granada, Avenida de la Fuente Nueva S/N, 18071 Granada, Spain; 5Instituto de Investigación Biosanitaria ibs.GRANADA, 18012 Granada, Spain

**Keywords:** acute myeloid leukemia, normal CD34^+^ bone marrow cells, hematopoiesis, differentiation, integrin, Toll-like receptor, patient heterogeneity, intracellular signaling, cellular communication, proteomics, mass spectrometry

## Abstract

Acute myeloid leukemia (AML) is an aggressive bone marrow malignancy that can be cured only by intensive chemotherapy possibly combined with allogeneic stem cell transplantation. We compared the pretreatment proteomic profiles of AML cells derived from 50 patients at the time of first diagnosis with normal CD34^+^ bone marrow cells. A comparison based on all AML and CD34^+^ normal cell populations identified 121 differentially abundant proteins that showed at least 2-fold differences, and these proteins included several markers of neutrophil differentiation (e.g., TLR2, the integrins ITGM and ITGX, and downstream mediators including RHO GTPase, S100A8, S100A9, S100A22). However, the expression of these 121 proteins varied between patients, and a subset of 28 patients was characterized by increased long-term AML-free survival, signs of myeloid AML cell differentiation, and favorable genetic abnormalities. These two main patient subsets (28 with differentiation versus 22 with fewer signs of differentiation) also differed with regard to the phosphorylation of 16 differentially abundant proteins. Furthermore, we also classified our patients based on their expression of 16 proteins involved in the regulation of iron metabolism/ferroptosis and showing differential expression when comparing AML cells and normal CD34+ cells. Among the 22 patients with less favorable prognosis, we could then identify a genetically heterogeneous subset characterized by adverse prognosis (i.e., death from primary resistance/relapse) and an iron metabolism/ferroptosis protein profile showing similarities with normal CD34^+^ cells. We conclude that proteomic profiles differ between AML and normal CD34^+^ cells; especially, proteomic differences reflecting differentiation and regulation of iron metabolism/ferroptosis are associated with risk of relapse after intensive conventional therapy.

## 1. Introduction

Acute myeloid leukemia (AML) is an aggressive malignancy characterized by the proliferation of immature leukemia cells in the bone marrow [1]. Its incidence increases with age [1,2,3], and the only potentially curative treatment is intensive chemotherapy possibly combined with allogeneic hematopoietic stem cell transplantation (allo-HSCT) [2]. However, such treatment is not possible for unfit patients and in the large group of elderly patients due to the unacceptable risk of severe or fatal toxicities [4,5,6]. Thus, there is a need for new therapeutic strategies to improve both the survival of younger patients with resistant/relapsed AML and elderly/unfit patients with a high risk of severe toxicity [2,4,5,6]. New targeted therapies are therefore tried in the treatment of AML [4,5,6,7,8,9].

AML is a heterogeneous malignancy with regard to AML cell morphology/differentiation, genetic abnormalities, and communication with neighboring nonleukemic cells in the bone marrow [1,2]. This heterogeneity is the basis for the subclassification of AML described in the WHO 2022 classification [1]. Despite the heterogeneity, there are also fundamental common biological characteristics, e.g., bone marrow homing, limited signs of AML cell differentiation, and rapid progression without treatment [1]. These last characteristics justify the therapeutic similarities between the various patient subsets when using conventional intensive chemotherapy as well as allo-HSCT [2]; the only exception being the acute promyelocytic (APL) variant [2].

The aim of this present study was to characterize the molecular consequences of AML transformation by comparing primary AML cells with normal CD34^+^ bone marrow cells, i.e., a normal counterpart for this bone marrow malignancy that also shows limited differentiation [10,11,12,13,14,15]. Furthermore, the acceptable intensity of AML therapy is determined by the balance between antileukemic efficiency versus the risk of severe/fatal toxicity, especially hematological toxicity, which is often dose-limiting and contributes to treatment-related mortality [2,4,16,17,18,19,20]. One hypothesis could then be that an unfavorable balance between anti-AML efficiency versus the risk of severe/lethal hematological toxicity for certain therapeutic approaches is due to biological similarities between normal and leukemic hematopoietic cells [20,21]. We would also expect proteomic differences between normal and leukemic hematopoietic cells to reflect fundamental AML cell characteristics and important mechanisms involved in leukemogenesis and possibly also chemoresistance against conventional as well as new targeted therapies [2,22,23]. Finally, proteomic profiling may become an alternative strategy for the evaluation of differentiation in AML cells. In this context, we performed a preliminary/pilot study where we investigated whether proteins showing differential expression when comparing normal and leukemic hematopoietic cells can be used for patient subclassification and the identification of patient subsets that differ in prognosis/survival after conventional intensive therapy. The present results suggest that our identification of distinct patient subsets is relevant for the prognostication of AML patients receiving conventional intensive chemotherapy, but this needs to be further investigated and confirmed in larger clinical studies.

## 2. Materials and Methods

### 2.1. Patient and Cell Samples

We reanalyzed a previous liquid chromatography with tandem mass spectrometry (LC–MS/MS)-characterized proteomic cohort including AML cells from 50 Caucasian patients with non-APL variants of AML at the time of first diagnosis (Table 1) [24]. This study included (i) all consecutive patients (acute promyelocytic leukemia being excluded) from the same geographical area during a defined time period (1995–2012) and receiving intensive conventional AML therapy (ii) with a high percentage of AML blasts among circulating leukocytes (see below) who (iii) completed the planned intensive and potentially curative antileukemic treatment. All patients had >20% AML cells among nucleated bone marrow cells. The patient cohort did not include patients with the uncommon erythroid and megakaryocytic AML variants.

None of the patients received autologous stem cell transplantation. Some patients received allogeneic stem cell transplantation according to national Norwegian guidelines; the time for allotransplantation is indicated in Appendix A.

The patients were classified according to the 2022 ELN risk classification by genetics at initial diagnosis [2] (Table 1). Only 26 patients were investigated with an extended panel of AML-associated mutations, and for this reason, 11 patients were classified as intermediate/adverse (i.e., absence of favorable genetic abnormalities) or unclassified (missing karyotyping or analysis of *NPM1/FLT3/CEBPA* genetic abnormalities).

Enriched primary AML cells could be prepared by density gradient separation alone from peripheral blood; all patients had ≥80% AML blasts among circulating leukocytes, and gradient-separated cells therefore included ≥95% AML blast cells. After gradient separation, the AML samples were cryopreserved according to a standardized protocol [25] and stored in liquid nitrogen until use. Briefly, cells were dissolved in standardized RPM1 1640 culture medium (Stem Cell Technologies, Vancouver, BC, Canada) supplemented with 40% heat-inactivated fetal calf serum (FCS) (Stem Cell Technologies) before being placed on ice; thereafter, an equal volume of cold medium with 20% dimethylsulfoxide (DMSO) was gradually added over 5 min to reach the final concentrations of FCS 20% and DMSO 10%. The cells were thereafter transferred to storage for 24–72 h at −70 °C before being transferred to liquid nitrogen. Cells were thawed at 0 °C and immediately transferred to the proteomic solution (see also Section 2.2) [24,26].

Cryopreserved normal CD34^+^ bone marrow cells were derived from eight healthy Caucasians (four men and four women; PromoCell GMBH; Heidelberg, Germany); their age did not differ from the AML patients (median age, 50.5 years; range, 41–55 years; interquartile range (IQR), 50.5 − 43.25 = 7.25). These cells were also cryopreserved in 10% DMSO. The proteomic analyses of the normal CD34^+^ cells have not been published previously but are available at ProteomeXchange, data identifier PXD058846.

### 2.2. Proteomic Analysis

Our methods have been described in detail previously [24,26,27], and we thus tested several sample preparation workflows [27]. These experiments included double versus single-digestion and in-solution versus filter-based digestions for proteome analysis, metal oxide affinity chromatography (MOAC), immobilized metal affinity chromatography (IMAC), and sequential elution from IMAC (SIMAC) for phosphoproteome analysis [28,29,30,31]. The results showed that FASP procedures produced the highest number of quantified proteins with a reduced number of missed cleavages. IMAC was selected because it produced the highest number of quantified phosphopeptides, even though MOAC, IMAC, and SIMAC protocols isolated different phosphoproteoforms.

Quantitative proteomics of the AML patients and CD34^+^ cells was performed according to the label-free quantification (LFQ) approach. Proteome extraction was carried out in 4% sodium dodecyl sulfate (SDS) and 0.1 M Tris-HCl (pH 7.6) with immediate boiling to inactivate proteases and phosphatases. Samples were reduced with 0.1 M of dithiothreitol (DTT) and alkylated with 50 mM iodacetamide (IAA) before being digested in a filter unit according to the filter-aided sample preparation (FASP) procedure, adding trypsin to 50 mM ammonium bicarbonate in a 1:50 ratio [32,33]. Quantitative phosphoproteomics was carried out with samples spiked with a super-SILAC (stable isotope labeling by amino acids in cell culture) mix [34]. The phosphoproteomics samples were FASP-processed and enriched for phosphopeptides using the IMAC approach [24]. LC–MS/MS runs were carried out with a Q Exactive HF Orbitrap mass spectrometer coupled to an Ultimate 3000 Rapid Separation liquid chromatography (LC) system. LC–MS/MS methods have been described in detail previously as Appendix A [24]. In brief, LC–MS/MS raw files for primary AML cells and normal CD34^+^ bone marrow cells were processed with MaxQuant software (version 1.5.2.8). The spectra were searched against the concatenated reverse-decoy Swiss-Prot Homo sapiens database (downloaded on 5 November 2015) utilizing the Andromeda search engine. Relative label-free quantification was performed using the MaxLFQ algorithm [35], with the LFQ count set to 1. This algorithm normalizes protein intensities based on peptide ratios measured in all pairwise comparisons within the entire sample batch. The MaxQuant parameters were configured as follows: cysteine carbamidomethylation was set as a fixed modification, while methionine oxidation, protein N-terminal acetylation, and Gln→pyro-Glu were included as variable modifications. Trypsin was stated as the digestion protease. The false discovery rate (FDR) was set at 0.01 for both peptides and proteins, and a minimum peptide length of six amino acids was required. Additionally, the match-between-runs and re-quantify options were enabled.

All proteomic raw data and MaxQuant output files together with the phosphoproteomic raw data can be found in the ProteomeXchange consortium with the dataset identifiers PXD014997 and PXD058846.

### 2.3. Morphological, Cell Surface Marker and Genetic Subclassification of AML Patients

The FAB classification is regarded as a standardized and well-described system to characterize and classify patients with regard to AML cell differentiation [36,37]. In our present study, we defined monocytic differentiation as FAB-M4/M5, neutrophil differentiation as FAB-M2, and undifferentiated as FAB-M0/M1. No erythroid or megakaryocytic variants (i.e., FAB-M6/M7 AMLs) were included.

Cell surface markers were analyzed with flow cytometry and the karyotype with standard cytogenetic analysis. Analyses of FLT3 and NPM1 mutations have been described previously [38]. Submicroscopic mutational profiling of 54 genes frequently mutated in AML was performed using the Illumina TruSight Myeloid Gene Panel and sequenced using the MiSeq system and reagent kit v3 (Illumina, San Diego, CA, USA) [38].

### 2.4. Statistical and Bioinformatical Analyses

Our various bioinformatical analyses are summarized in the flow chart in Figure 1.

The Perseus 2.0.7.0 platform was used for functional and statistical analyses of proteomics and phosphoproteomics data [39]. Patient subset data were normalized by using width adjustment. A Welch’s *t*-test with Benjamini-Hochberg correction (i.e., FDR < 0.01) was performed to test for significant difference between means of compared groups. Z-statistics were carried out on the Welch’s *t*-test regulated proteins to identify proteins with significantly different fold changes [40]. A flow chart summarizing the various bioinformatical analyses and their presentation in our study is given in Figure 1.

Welch’s *t*-test with *p*-value < 0.05 was performed to test for significantly differentiated phosphorylation sites between compared groups. Reactome pathway, Gene Ontology (GO), and KEGG pathway enrichment analyses were obtained with the Enrichr gene set search engine [41,42,43,44]. Protein–protein interaction (PPI) network analyses were performed with the STRING database version 11.5. Networks were visualized using the Cytoscape platform v3.10.0 [45]. GraphPad v8.0.1 and the Perseus platform were used to make the volcano plot and the heatmap, respectively.

## 3. Results

### 3.1. Primary AML Cells at the Initial Diagnosis Show Increased Abundance of Several Neutrophil Differentiation Biomarkers Compared with Normal CD34^+^ Bone Marrow Cells; Results from an Initial Statistical Comparison

The total proteomic profiles for all 50 AML patients and eight normal CD34^+^ bone marrow cell populations were compared. We first used Welch’s *t*-test with Benjamini–Hochberg correction (i.e., FDR < 0.01) based on only those proteins with at least 70% valid values both for leukemic and normal cells. A total of 891 proteins showed a statistically significant difference (i.e., *p* < 0.05); this included 184 proteins with at least a 2-fold median increase for the AML cells (Appendix A) and 199 proteins showing at least a 2-fold increase for the CD34^+^ normal bone marrow cells (Appendix A).

The 10 top-ranked Reactome terms for the 184 significant proteins with at least 2-fold increased levels in AML cells are listed in Appendix A together with the corresponding differentially abundant proteins for each of these terms. The terms reflecting neutrophil differentiation/innate immunity included a relatively large number of overlapping proteins, whereas fewer proteins were included in terms reflecting integrin function, endothelial cell function, and RHO signaling, respectively. There was a considerable overlap of proteins between each of these last minor groups and the terms reflecting neutrophil functions/innate immunity. These top 10 Reactome terms included 79 of the 184 proteins with increased levels in the AML cells.

The top-ranked Reactome terms for the 199 proteins with an at least 2-fold increase in normal CD34^+^ bone marrow cells reflected nucleotide metabolism/synthesis (purine ribonucleotide monophosphate biosynthesis, metabolism of nucleotides, nucleotide biosynthesis), regulation of DNA functions/mitosis (DNA strand elongation, activation of pre-replicative complex, mitotic G1 phase, and G1/S transition), and integrin signaling (p130CAS linkage to MAPK signaling for integrins, BRB2:SOS provides linkage to MAPK signaling for integrins) (Appendix A). However, these terms included only 33 of the 199 proteins with high levels in the CD34^+^ cells.

We conclude that the differentially abundant protein identified in this initial comparison of AML and normal CD34^+^ cells shows extensive biological diversity. The main difference was that proteins with increased levels in AML cells mainly reflected limited myeloid/neutrophil differentiation, whereas proteins with increased levels in normal CD34^+^ cells reflected a different regulation of cellular proliferation and integrin-associated intracellular MAPK signaling.

### 3.2. Proteomic Comparison of AML Cells at Initial Diagnosis and Normal CD34^+^ Bone Marrow Cells: High Expression of Neutrophil Markers in the AML Cells and High Levels of Platelet/Coagulation Biomarkers in Normal Cells

Our initial strategy was the comparison of the protein levels for all 50 AML patients and the eight normal CD34^+^ bone marrow cell populations by using Welch’s *t*-test with Benjamini–Hochberg correction (i.e., FDR < 0.01) as described in Section 3.1. As an alternative and additional statistical strategy to our previous use of a fold change cut-off value (i.e., 2-fold; see Section 3.1) we conducted a significant fold change difference analysis (*Z*-statistics) of the 891 proteins identified in the initial step analysis described in Section 3.1. By using this approach, we identified only 48 proteins with significantly increased levels in AML cells and 73 proteins with increased levels in the normal CD34^+^ bone marrow cell populations (Appendix A; see also Figure 2); all these proteins were among the most significant in Welch’s *t*-test with Benjamini–Hochberg correction (as indicated in Appendix A).

Reactome analysis (Figure 2A) of the 48 AML-associated proteins showed enrichment of proteins reflecting neutrophil degranulation (21 proteins) and innate immunity (17 overlapping proteins). The cellular compartment (CC) GO terms also reflected differences consistent with neutrophil differentiation, i.e., granular/vesicular lumen. Furthermore, the top four Reactome terms for the AML cells also included the terms RHO GTPases activate NADPH oxidases (four proteins) and antimicrobial peptides (five proteins) (Figure 3). The top-ranked terms from the other GO term analyses (biological process, BP; molecular function, MF) and the KEGG analysis showed lower significance than the Reactome/CC terms (Figure 2A).

The analyses of normal CD34^+^ bone marrow cells reflected increased levels of a relatively small number of proteins involved in coagulation/platelet activation; the four top-ranked Reactome terms included a total of six proteins out of the seventy-three significantly increased proteins in CD34^+^ cells (Figure 3). The terms identified by GO, Reactome, and KEGG analyses of these 73 proteins showed generally lower significance than the Reactome/CC terms for the AML cells (Figure 2).

Based on these analyses, our conclusion is the same as for the analysis described after the first statistical analysis described in Section 3.1—differentially abundant proteins between AML and normal CD34^+^ cells show large biological diversity with the main characteristic being an increased expression of proteins reflecting limited myeloid/neutrophil differentiation of the AML cells, whereas the increased levels of regulators of the proliferation/cell cycle possibly reflect the immature/stem cell status of the CD34^+^ cells.

### 3.3. Analysis of Differentially Abundant Proteins by Volcano Plot Analysis; Identification of Neutrophil Differentiation Markers as Well as Regulators of Cellular Metabolism Including Iron Metabolism

We conducted a volcano plot analysis of the 121 differentially abundant proteins identified after the fold change significance analysis (Figure 2B). Only ten biologically diverse proteins (five increased in AML and five in CD34^+^ cells) showed a marked differential expression, including myeloid differentiation markers (HBA1, HBB, PRG2, and possibly the antimicrobial PRGT); regulators of lipid (ALDH1A1, PLBD1), amino acid (PHGDH, possibly APCS), and iron (TF) metabolism; and transcriptional regulators (HIST1H1E, possibly APCS) (Appendix A). Thus, this analysis also identified several differentiation markers but, in addition, a few metabolic and functional DNA/transcriptional regulators.

### 3.4. Analysis of Protein–Protein Interaction Networks; Identified Networks Involve Regulators of Intracellular Signaling, Platelet Function, and Iron Metabolism, as Well as Transcription and DNA Repair

Protein–protein interaction (PPI) network analyses based on the 121 differentially abundant proteins from the fold change significance analysis identified only four networks with at least three members (Figure 4):*Regulation of MAPK cascade.* This network included nine proteins, three of them showing increased levels in primary AML cells. Differences in regulators of integrin-associated MAPK regulators were also identified in our first global analysis (see first Section 3.1 of Results and Appendix A, right columns).*Nitric oxide transport.* The network included the two hemoglobin chains also identified in the volcano plot together with the hemoglobin-binding protein haptoglobin.*Base excision repair.* This network included three proteins with decreased levels in AML cells.*Aldehyde dehydrogenase.* This network included three proteins with decreased levels in AML cells.

Thus, only one relatively large network was identified, suggesting that the intracellular molecular context of MAPK signaling is different for primary AML cells compared with normal CD34^+^ bone marrow cells. Only 18 out of 121 differentially abundant proteins were included in these four networks.

We identified seven additional pairs of interacting proteins. Three pairs included proteins that were increased in the AML cells, including (i) one pair with two integrin alpha chains and (ii) two pairs with three proteins (S100A8, S100A9, and AHNAK) involved in signaling/calcium metabolism. Furthermore, four pairs were decreased in AML cells, and these are referred to as platelet-associated proteins (GP1BA and GP1BB), iron metabolism (transferrin and its receptor), cell cycle/histone regulation (CDK1, H1F0), and two members of the family structural maintenance of chromosomes (SMC) that modulate chromosome structure during mitosis (SMC2, SMC4). However, only 33 of the 121 proteins were included in interacting networks/pairs, an observation suggesting that the identified 121 proteins show a considerable biological heterogeneity.

### 3.5. Even Proteins Showing Strong Differential Abundance When Comparing AML and CD34^+^ Cells, in General, Vary Between Individual Patients and Can Be a Basis to Identify AML Patient Subsets

We conducted an unsupervised hierarchical clustering analysis based on the 121 differentially abundant proteins identified after *Z*-statistics (Figure 5). The differentially abundant proteins were separated into two main clusters (see left part of the figure, referred to as platelet activation and neutrophil degranulation, respectively), and both main clusters could be further subdivided into two subclusters. The upper main cluster included those proteins that reflected platelet activation/function in the Reactome analysis (Figure 2A; Appendix A), and all these proteins were located in the lower subcluster (Appendix A). On the other hand, the lower main protein cluster included the large majority of proteins reflecting the terms neutrophil degranulation/innate immune system, RHO GTPases activate NADPH oxidases and antimicrobial peptides in the Reactome analysis (Figure 2B, Appendix A), and the large majority of these proteins were included in the lower subcluster (Appendix A).

The normal CD34^+^ bone marrow cell samples formed a separate main cluster in this analysis; see the top of Figure 5 to the right. The primary AML cell samples formed the large left main cluster that could be further subdivided into several subclusters. Furthermore, this analysis showed that patients could be divided into two main subsets based on the expression of a subset of proteins showing generally high levels when comparing the total AML and CD34^+^ cell populations (the lower protein main cluster; see left part of Figure 5). The 22 patients on the left showed a generally lower expression of these proteins compared with a higher expression in the 28 patients on the right.

The clinical and biological characteristics of these two patient subsets are presented in Appendix A and are compared/summarized in Appendix A. The 28-patient subset on the right had a significantly lower age (Appendix A; Fisher’s exact test, *p* = 0.00076), a lower frequency of undifferentiated FAB-M0/M1 leukemic cells (*p* = 0.0003), a lower frequency of CD34^+^ AML cell populations (*p* = 0.0307), higher frequencies of favorable (*p* = 0.0312) and favorable/normal karyotypes (*p* = 0.0299), a lower frequency of refractory/relapsed disease (*p* = 0.0365), and a higher frequency of long-term AML-free survival (*p* = 0.0016) compared with the 22-patient subset on the left. These associations are mainly caused by several patients with a favorable karyotype (i.e., inv(16)) who clustered close to each other among the 28-patient subset on the right.

We compared the two clusters with regard to peripheral blood blast count and the percentage of AML blast cells among the nucleated bone marrow cells. The 28 patients in the neutrophil degranulation/differentiation cluster showed a significantly higher peripheral blood blast count (median 42.5 × 10^9^/L, range 4–357 × 10^9^/L, IQR 71.0 − 29.5 = 41.5 × 10^9^/L) than the other 22 patients (median 23 × 10^9^/L, range 5–231 × 10^9^/L, IQR 47.25 − 10.75 = 36.5 × 10^9^/L; Mann–Whitney U test, *p* = 0.026). In contrast, the percentage of bone marrow blasts did not differ between the 28 neutrophil differentiation patients (median 76.5%, range 30–97%, IQR 85.75 − 60.50 = 25.25) and the other 22 patients (median 83.5%, range 25–99%, IQR 95.00 − 58.25 = 35.25).

Cryopreservation and storage in liquid nitrogen can lead to a reduced abundance of membrane molecules [46]. Several of the identified neutrophil differentiation markers included in the clustering analysis are cell surface molecules. We, therefore, compared the cell storage times for the left 22-patient main cluster (median 86 months, range 60–240 months, IQR 139 − 66 = 73 months) and the right 28-patient main cluster with the neutrophil differentiation patients (median 131 months, range 60–245 months, IQR 225.00 − 72.25 = 152.25 months). This difference was not statistically significant.

### 3.6. Additional Phosphoproteomic Differences of the 121 Differentially Abundant Proteins from the AML/CD34^+^ Cell Comparison: A Comparison of the Two Patient Subsets Identified in the Proteomic Clustering Analysis of These 121 Proteins

Primary AML cells derived from 41 of the 50 patients were included in a previous phosphoproteomic study [24]; 23 of these patients belonged to the larger main subset (Figure 5, the 28 patients on the right) and the other 18 patients belonged to the smaller main subset (Figure 5, the 22 patients on the left). We observed detectable phosphorylation for 38 of the 121 differentially abundant proteins identified in our proteomic comparison of primary AML cells and normal CD34^+^ bone marrow cells (i.e., proteins with a fold difference exceeding 2.0; see above). These 38 proteins had 174 detectable phosphorylated sites. Next, we conducted a statistical comparison (Welch’s *t*-test) of the phosphorylation level for these 174 phosphosites/38 differentially abundant proteins between the two patient subsets identified in the unsupervised hierarchical clustering analysis presented in Figure 5 (i.e., 23 of the 28 right patients versus 18 of the 22 left patients; see first chapter of this section). This analysis identified a total of 53 differentially phosphorylated sites in 16 proteins that differed significantly between these two patient subsets. These differing phosphosites are presented in Table 2 and Appendix A; important characteristics of the 16 phosphoproteins are presented in Appendix A.

Based on the data presented in Table 2 and Appendix A, the following main observations were made:Three of these 16 proteins are regarded as markers of neutrophil differentiation (ANXA2, ITPR1, and PRKCD; see Figure 3).For seven of these sixteen proteins, differences were detected for at least two phosphosites, i.e., the phosphorylation profile differed and not only in the phosphorylation of a single site.For certain proteins, opposite differences/effects on various phosphosites were observed, i.e., AHNAK, PLEC, and TOP2A (see Appendix A).For other proteins, the total protein level was increased/decreased in one direction between the two patient subsets, whereas the level of (certain) phosphorylations was altered in an opposite direction (AHNAK, ITPR, PBXIP1, PLEC, TOP2A); these observations show that altered phosphorylation reflects the altered regulation of phosphorylation and not the difference in total protein level.Some of these phosphosites seem important for molecular compartmentalization (AHNAK) or molecular interactions/intracellular signaling (AHNAK, ANXA2, GP1BB, ITPR1, PLEC, PRKCD, STMN1).The phosphorylation status also seems important for the regulation of fundamental cellular processes, especially cell cycle regulation/cell growth (ANXA2, KCTD12, PLEC, PRKCD, SMC4, TOP2A) but also stemness/stem cell differentiation (AHNAK, STMN1), survival/chemosensitivity (ANXA2, DUT, PRKCD, TOP2A), cytoskeleton functions (DBN1, PLEC), electrolyte balance/transport (DBN1, ITPR1), nucleotide metabolism (DUT), cell migration (AHNAK), nutrition (APOBR), and epigenetic/transcriptional regulation (MSL1, PBXIP1, RCOR3, TOP2A).Some of the proteins seem to be implicated in carcinogenesis (KCTD12, MSL1, PBXIP1, PRKCD, STMN1) and even AML chemosensitivity (ANXA2, STMN1, TOP2A).

To conclude, primary AML cells and normal CD34^+^ bone marrow cells show complex differences in their proteomic profiles, and this complexity is further increased by additional variations between patient subsets with regard to post-translational modulation/phosphorylation for 16 of the 121 differentially abundant proteins.

### 3.7. Variation of the Biological Context of Transferrin and Transferrin Receptors That Are Both Differentially Abundant When Comparing Normal and Leukemic Hematopoietic Cells; Identification of a Patient Minority Whose AML Cells Show Similarities with Normal Cells in Regulation of Cellular Iron Utake

Our statistical comparison of primary AML cells and normal CD34^+^ bone marrow cells showed that both transferrin and the transferrin receptor were differentially abundant with low levels in the leukemic cells; transferrin was then the protein with the lowest *p*-value and its receptor also showed a highly significant decrease (Figure 4, Appendix A). Transferrin and its receptor constitute the main mechanism for the cellular uptake of iron [77,78,79]. Furthermore, iron metabolism and the regulation of iron-dependent ferroptosis seem to have a prognostic impact on human AML [80,81,82,83,84,85,86,87,88]. For these reasons, we compared iron metabolism in AML cells and normal CD34^+^ bone marrow cells in more detail.

The Reactome term “Iron metabolism” and the KEGG term “Ferroptosis” included only three of the 121 differentially abundant proteins showing at least a 2-fold difference when comparing AML cells and normal CD34^+^ bone marrow cells and, in addition, being identified in the z-score analysis (Appendix A). However, 13 additional iron metabolism/ferroptosis proteins also showed statistically significant (Welch’s *t*-test with Benjamini–Hochberg correction) but without being identified in the z-score analysis (see figure legend). Our comparison of iron metabolism/ferroptosis regulation in primary AML cells and normal CD34^+^ bone marrow cells was, therefore, based on these 16 proteins (Appendix A). The identified proteins are involved in the regulation of cellular iron uptake and intracellular endosomal release (TFRC, TF, ACO1, several V-ATPase components), regulation of iron homeostasis (ACO1), and/or various steps/characteristics of ferroptosis—i.e., altered regulation of cellular iron metabolism/handling (TFR, TFRC, ACO1, CUL1, CAND1), altered redox balance (GLRX3), altered lipid metabolism (ACSL1), and protein ubiquitination/degradation (RPS27A, CAND1, CUL1, NED8, ATG7) (Appendix A, [80,81,82,83,84,85,86,87,88,89,90,91,92,93,94]).

We conducted an unsupervised hierarchical clustering analysis based on the 16 differentially abundant metabolic/ferroptotic proteins identified as described above (Figure 6). The proteins are listed to the left in the figure. Only three proteins, TF, TFRC (both increased in normal cells), and ATP6V0D1 (higher in AML cells), showed ≥2-fold differences when comparing leukemic and normal cells, whereas the other 13 differentially abundant proteins showed <2-fold differences that still reached statistical significance (*p* < 0.05) when comparing the two complete groups. It can be seen that the left main cluster included a majority of 40 AML patients (indicated by blue color), whereas the right minor main cluster had one right subcluster that included the eight normal CD34^+^ cell populations (indicated by corn silk color at the top of the figure) and another subcluster that included 10 AML patients (blue color).

It can be seen from Figure 6 that the minority of 10 patients clustering together with the normal CD34^+^ bone marrow cells differed in their proteomic profile both from the normal cells (although the common clustering reflects certain similarities) and from the 40 other AML cell populations. Thus, we identified a minority of 10 AML patients whose leukemic cells showed a different iron metabolism/ferroptosis protein profile compared with cells derived from the other 40 patients and from the eight normal individuals. The similarities between the 10 exceptional AML patients and the healthy individuals especially included the absorption pair transferrin/transferrin receptor and V-ATPase components that are important for vacuolar acidification and thereby intracellular iron release [77,78,79].

We compared the clinical and biological characteristics of this minority of 10 AML patients (i.e., the right AML subcluster) with the other 40 patients (the left main cluster). First, all 10 of the exceptional patients had high-risk/chemoresistant AML (i.e., primary resistance or death from later resistant relapse), whereas 18 of the 40 other patients reached long-term AML-free survival (Fisher’s exact test, *p* = 0.0085). Second, the minority of 10 patients (median age 62 years; range 29–80 years; IQR 67.25 − 57.50 = 9.75) had a higher age than the 40 patients in the left main cluster (median 46 years; range 18–67 years; IQR 57.00 − 37.25 = 19.75; Mann–Whitney U-test, *p* = 0.00214). Finally, the 10 AML patients in the right main cluster and the 40 other patients in the left main cluster did not differ significantly with regard to:Male versus female ratio.A low frequency of secondary AML.Morphological signs of differentiation, i.e., frequency of monocytic FAB-M4/M5 AML variants (Fisher’s exact test, *p* = 0.1548).Number of patients with at least 20% AML cells expressing the CD34 stem cell marker (*p* = 0.0706).Even though all six patients with the favorable inv(16) karyotype were localized in various subclusters among the left 40 patients, this difference did not reach statistical significance.Other cytogenetic abnormalities did not differ significantly between the two main AML patient subsets; both subsets showed heterogeneity with regard to karyotype.The frequencies of *FLT3-ITD* or *NPM1-INS* did not differ.Both groups were heterogeneous with regard to the ELN risk classification by genetics at the time of initial diagnosis [2] (see Table 1 and Appendix A). The minority of 10 patients in the right main cluster included three patients with ELN adverse prognosis, four patients with intermediate/adverse prognosis, and one patient for each of the three groups: favorable, intermediate, and unclassified.The peripheral blood blast count of the 10 patients that were clustered together with normal CD34^+^ cells (median 26.0 × 10^9^/L, range 5–71 × 10^9^/L, IQR 49.75 − 9.75 = 40.0 × 10^9^/L) did not differ significantly from the other 40 AML patients (median 33.5 × 10^9^/L, range 4.0–351 × 10^9^/L, IQR 99.5 − 27.5 = 72.0 × 10^9^/L). Similarly, the percent AML blasts in the bone marrow did not differ between the minor subset of 10 patients (median 63.5%, range 25–99%, IQR 95.00 − 49.25 = 45.75%) and the majority of 40 patients (median 83.5%, range 25–99%, IQR 86 − 53 = 33) in the other main cluster. These observations suggest that there is no major difference between the two patient subsets with regard to AML cell burden.All 10 patients clustering together with the normal CD34^+^ cells belonged to the left 22-patient cluster/subset identified in Figure 5 (see Appendix A). Thus, in the 121-protein clustering a subset of 22 patients was identified that showed few signs of differentiation and adverse prognosis (i.e., decreased frequency of long-term survival), and in this second clustering analysis (Figure 6) based on regulators of iron metabolism/ferroptosis, a subset among these 22 patients (i.e., patients with fewer signs of differentiation) was identified with a particularly high-risk disease/adverse prognosis.

To summarize, our iron metabolic/ferroptotic clustering analysis based on these 16 differentially abundant proteins identified a heterogeneous minority of elderly patients with chemoresistant AML; these patients were heterogeneous both with regard to AML cell differentiation, CD34 expression, karyotype, *FLT3-ITD*, and *NPM1-INS*. Thus, the significant difference in long-term AML-free survival between these two patient subsets cannot be explained by different frequencies of other prognostic parameters.

## 4. Discussion

We compared the proteomic profiles for primary human AML cells and normal CD34^+^ bone marrow cells. Even though the two groups differed with regard to the expression of several neutrophil markers and regulators of iron metabolism/ferroptosis, the patients were heterogeneous even with regard to the expression of these markers.

### 4.1. Methodological Considerations

Our study included 50 consecutive patients receiving intensive AML therapy; they came from a defined geographical area during a defined time period [24]. Our study should, therefore, be regarded as a population-based study of AML patients with a high percentage of AML cells among circulating leukocytes. The morphological classification of differentiation was performed according to the FAB/WHO 2016 classification that represents a standardized system describing detailed and updated morphological/biological/diagnostic criteria for the subset “AML not otherwise specified”, including how various subsets in this class correspond to the previous FAB classification [95].

We also classified our patients according to the ELN risk classification by genetics at initial diagnosis (Table 1). Because the genetic analyses were incomplete for several early patients and additional biological material was not available, twelve patients were classified as intermediate/adverse and six patients could not be classified. We would expect most of the intermediate/adverse patients to have an intermediate risk [2].

Our study did not include patients with the uncommon erythroid (FAB-M6) and megakaryocytic (FAB-M7) variants [1,95]. The higher levels of platelet/megakaryocyte markers in the normal CD34^+^ cells probably reflect the heterogeneity of CD34^+^ cells including immature erythroid/megakaryocytic stages [11,13,14,15,96].

The use of gradient-separated AML cells from peripheral blood in our study has been discussed in detail in a previous study. The reasons for this methodological approach and its possible limitations are as follows [24]:As discussed in previous methodological publications, more extensive cell separation procedures can alter the functional characteristics of AML cells [97,98]; by selecting patients with a high percentage of circulating AML blasts, we could prepare highly enriched AML cell populations by using simple gradient separation alone. This would not be possible with relatively low but still diagnostic percentages of bone marrow blasts.A previous methodological study showed that cryopreservation had only a limited influence on the proteomic profiles of AML cells [34].The AML cell population has a hierarchical organization, and leukemic stem cells are possibly responsible for how mortality relates to AML chemoresistance [98], but despite this, we investigated the whole AML cell population. However, several previous studies have also demonstrated that biological characteristics of the whole AML cell population reflect the risk of resistance after chemotherapy, e.g., cytokine release, gene expression, and epigenetic and pathway activation profiles (for reference, see ref. [24].Previous studies have shown that AML patients with and without peripheral blood leukemization have comparable frequencies of various genetic abnormalities [99], i.e., there is no strong enrichment of certain (high-risk) abnormalities when using our strategy for inclusion of patients.Previous studies suggest that leukemization may have an adverse prognostic impact, and if so, our inclusion of only patients with leukemization may cause an enrichment of patients with a more aggressive disease compared with AML in general. However, studies of the possible independent prognostic impact of high peripheral blood AML blast counts have given conflicting results [100,101]; if an effect is present, it must be weak [102] and/or may be present only for certain minor patient subsets [100] and/or be present only when counts exceed 50–100 × 10^9^/L [100,103,104] (most of our patients had lower counts).Finally, all our patients had >20% blasts in the bone marrow, whereas certain forms of AML can now be diagnosed by lower blast percentages in bone marrow or peripheral blood [1].

Even though it can be argued that our observations are relevant for AML in general, our results should be interpreted with great care, and, due to our selection of patients with leukemization, they may be representative only for patients with >20% blasts in the bone marrow and/or peripheral blood leukemization.

### 4.2. Increased Neutrophil Differentiation of Primary AML Ells Compared with Normal CD34^+^ Bone Marrow Cells

We identified 121 differentially abundant proteins that should be regarded as markers of neutrophil differentiation (Figure 2A Reactome terms, Appendix A). However, the low number of proteins included, the lower statistical significance of other GO/Reactome/KEGG terms (Figure 2A), the results from the volcano plot analysis (Figure 2B), and the relatively few and small identified protein interaction networks (Figure 4) possibly reflect a biological heterogeneity of those differentially abundant proteins not associated with neutrophil differentiation. However, the differentially abundant proteins included cross-communicating cell surface molecules and their downstream mediators:*TLR2 and TLR2 signaling in AML.* Primary AML cells express functional TLRs, including TLR2 and TLR4 [105]. Both of these receptors can bind endogenous agonists [106], although these observations have been questioned and may at least partly be due to a function as assistants due to their binding of other ligands [107]. The transmembrane TLR2 protein was increased in our AML cells (Figure 2). The downstream TLR2 signaling includes MYD88, various IRAKs, TAK1, TAB1/2, and TRF6 adaptor molecules; IRAK-6 thus seems responsible for recruitment to the receptor complex that moves to the cytoplasm and activates NFκB and the AP-1 transcription factor [108]. This MYD88-dependent downstream signaling is not specific for TLR2 but seems common for various TLRs, including TLR4.*Alternative signaling downstream to TLR2.* Alternative TLR2/MyD88 downstream targets are PI3K-Akt with the enhancement of ERK1/2, p38 MAPK, JNK1/2, and focal adhesion kinase (FAK) activation/phosphorylation [106]. Many of these targets are also important for the downstream signaling of TLR4 [108], a receptor that interacts with S100A9 (see below) and thereby regulates AML cell differentiation by targeting p38 and ERK1/2 [109,110]. Thus, there are functional links between TLR2, TLR4, and S100A9/S100A10.*Integrins and integrin signaling including RHO GTPases.* Our study suggests that the integrin expression profile is altered in AML cells, especially ITGAX, ITGAL, and ITGA2B (Figure 2, Appendix A). Integrins bind a wide range of cell adhesion and extracellular matrix molecules, and their outside-in signaling modulates the activation of several intracellular mediators including FAK, PI3K, AKT, mTOR, ERK, and JNK [111,112,113,114] as well as RHO-GTPases that are also involved in inside-out signaling and the regulation of the proliferation, differentiation and possibly chemosensitivity of malignant cells, including AML cells [111,115,116,117,118,119,120]. Thus, integrins share downstream target TLR2 and/or S100 proteins (see below).*S100 proteins in AML.* Our results suggest that S100A8/9/11, especially, shows increased levels in primary AML cells. Both S100A8 and S100A9 are regarded as differentiation markers in AML but also seem to limit/inhibit further AML cell differentiation [121]. A high mRNA expression of S100A8 is associated with adverse prognosis in AML with a normal karyotype, and this prognostic impact may depend on its effects on autophagy, the production of reactive oxygen species, and the mitochondrial regulation of apoptosis [121,122]. Furthermore, S100A8 and S100A9 exist both as homodimers and heterodimers, and the effect of the heterodimer is difficult to predict. Thus, there is a crosstalk both between S100A8 and S100A9 [121] and also between S100A8 and TLR4 [109]. Finally, S100A11 is also a regulator of cellular proliferation and seems to have a prognostic impact on certain malignancies, but this is the first report to suggest a role in human AML [123].*Fibrinogen.* Fibrinogen is a hexameric glycoprotein that consists of two sets of three α, β, or γ chains [124]; it is mainly synthesized in hepatocytes but can also be abundant in certain other cells [125]. The expression of all the fibrinogen α, β, and γ chains was generally higher for normal CD34^+^ bone marrow cells than AML cells (Figure 2, Appendix A). Fibrinogen can bind to αMβ2 and αXβ2 integrins [126] as well as soluble ferritin [127], whose systemic levels can be increased and predict adverse prognosis in AML [128]. These observations further support our hypothesis that AML cells and normal CD34^+^ bone marrow cells differ with regard to integrin function but also iron metabolism (Figure 4) due to an increased expression of transferrin–transferrin receptors in CD34^+^ cells.

Taken together, these observations suggest that the differentially abundant proteins (Appendix A) include TLR2 and several integrins together with certain common and interacting downstream mediators. Several of the proteins are regarded as potential therapeutic targets in AML, including the integrin system [116,117,129] with the downstream MAP kinases [130,131,132,133,134,135,136], S100A molecules [121,122,137,138], and TLR2/4 with their downstream mediators [106,108], including the RHO GTPases [111,115,116,117,118,119,120,139,140]. Finally, aldehyde dehydrogenase is also regarded as a possible therapeutic target in AML [141,142,143,144], and the biological context of this enzyme seemed to differ between AML cells and normal CD34^+^ cells.

Future studies have to clarify whether the high levels of possible therapeutic targets in our present study represent a high dependency on these mediators and, thereby, susceptibility to targeted therapy, or whether resistance to targeting due to the high levels requires high inhibitor concentrations for antileukemic efficiency. Furthermore, the therapeutic targeting of these differentially abundant mediators would probably also influence the function of various immunocompetent cells known to infiltrate the leukemic bone marrow microenvironment [145,146]; they may thereby influence the balance between antileukemic and AML-supporting immune activity. Thus, the therapeutic targeting of these mechanisms may have direct effects on AML cells and additional indirect effects via neighboring immunocompetent or bone marrow mesenchymal/stromal cells [108].

Previous studies have investigated the possible prognostic impact of differentiation markers in AML, including CD34 expression and Sudan black staining; these early studies suggested that such markers had no/uncertain prognostic impact [147,148], whereas certain recent studies suggest that CD34 expression is important in certain AML subsets [149,150]. Furthermore, minimal residual disease has now become important for clinical prognostication in AML [151,152,153], and the pretreatment burden of immature/undifferentiated cells (i.e., the degree of differentiation within the AML cell population) can be combined with MRD evaluation to improve prognostication [154]. In this context, we conclude/suggest that the evaluation of AML cell differentiation with proteomic profiling rather than the use of single or a limited number of differentiation markers, and possibly in combination with MRD estimation, should be regarded as a possible strategy for prognostication in AML. Future clinical studies then have to clarify whether other patients with a similar differentiation marker profile as described for patients with a favorable karyotype in our present study (Figure 3, right patient subset; see also Appendix A) also have a similar favorable prognosis.

We identified 121 proteins that showed differential expression when comparing all 50 primary AML cell populations with the normal CD34^+^ bone marrow stem/progenitor cells. However, even the expression of these differentially abundant proteins showed a considerable variation between individual patients with regard to protein levels, and these markers could therefore be used for further subclassification of our 50 patients into two main subsets (see Figure 5). Furthermore, our phosphoproteomic studies illustrate that the proteomic complexity is further increased by differences in the degree of post-translational phosphorylation between the two main patient subsets for 16 of the 121 differentially abundant proteins (see Section 3.6). Previous studies suggest that these phosphoproteomic differences (see Table 2 and Appendix A) are functionally important, although it should be emphasized that most of these functional studies have not been performed in normal or malignant hematopoietic cells [94,106,107,108,109,110,111,112,113,114,115,116,117,118,119,120,121,122,123,124,125,126,127,128,129,130,131,132,133,134,135,136].

We conducted a clustering analysis based on the 121 proteins, showing at least a two-fold difference when comparing AML cells and CD34^+^ bone marrow cells (Appendix A), and this analysis identified a subset of 28 patients characterized by neutrophil differentiation and favorable prognosis (Appendix A for details). These 28 patients were also characterized by higher peripheral blood blast counts than the other patients, whereas the bone marrow blast percentages did not differ. These observations are consistent with the hypothesis that the degree of peripheral blood leukemization is mainly determined by the biological characteristics of the AML cells rather than the AML cell burden. Furthermore, this hypothesis is further supported by the previous observation that hyperleukocytosis is also more common in patients with myelomonocytic AML cells [155,156]. Finally, it is not unexpected that AML cells sharing biological characteristics with a mature normal myeloid cell type that is determined to enter the circulation also show a similar and higher degree of migration/leukemization compared with more undifferentiated ML cells.

### 4.3. Iron Metabolism and Regulation of Ferroptosis in Primary AML Cells; Comparison with Normal CD34^+^ Bone Marrow Cells and the Variation Between Patients

Ferroptosis is a form of programmed cell death characterized by altered iron metabolism together with the formation of reactive oxygen species and altered lipid metabolism; it should thus be regarded as iron-dependent cell death [80,81,82,83,84,85,86,87,88]. We observed several differences between AML cells and normal CD34^+^ bone marrow cells with regard to iron uptake. First, decreased AML cell levels of the transferrin receptor suggest decreased uptake of iron-containing transferrin, and decreased receptor-mediated transferrin uptake is possibly reflected by the decreased AML cell transferrin levels. Second, V-ATPase is important for endosomal acidification, which is essential for intracellular iron release after endosomal uptake both in the cytoplasm and possibly also in mitochondria [79]. In our clustering analysis based on the 16 differentially abundant proteins involved in iron metabolism/ferroptosis (Figure 4), we identified a patient subset characterized by adverse prognosis (i.e., primary chemoresistance or later relapse) that clustered together with the normal CD34^+^ cells and showed relatively low levels of most iron metabolism/ferroptosis proteins. These observations are also consistent with a previous study describing increased mRNA levels of V-ATPase components and different iron metabolism for primary AML cells derived from patients with favorable prognosis [157], as well as another study describing an association between increased relapse risk and decreased protein levels of V-ATPase components [24]. Finally, in contrast to the patients identified based on the clustering analysis of all highly differing proteins (i.e., more than 2-fold and including several differentiation markers; see Figure 2), the patient subsets with adverse prognosis identified in the clustering analysis of iron/ferroptosis markers (Figure 3) did not differ from the other patients with regard to age, morphological AML cell differentiation, AML cell expression of the CD34 stem cell marker, or genetic abnormalities. Taken together, these observations suggest that the regulation of iron metabolism/ferroptosis should be further investigated as a possible independent prognostic marker for AML patients receiving intensive conventional therapy.

### 4.4. Limitations of This Present Study

The complexity of cellular proteins far exceeds that of the approximately 20,000 protein-encoding genes in humans [158]. The term proteoform is therefore used to describe the complexity of the protein system [158,159,160], and it is defined as all the different molecular forms in which the protein product of a single gene can be found [159]. This variation is created by several mechanisms and includes (i) the protein-encoding gene and its genetic variants, (ii) the various alternatives for RNA splicing, and (iii) variations in folding/function compared with the canonical protein due to genetics/amino acid sequence differences, splice variants, posttranslational modification (e.g., phosphorylation, methylation, acetylation, glycosylation, lipidation), and protein truncation (e.g., activation by proteolytic cleavage) [158,159,160,161,162,163,164,165]. Our present study has only investigated a part of this complexity, i.e., an estimate of the total protein level and for the differentiation studies of posttranslational phosphorylation. We did not, for example, investigate splicing variants (also referred to as isoforms in the previous literature and the gene database) that are important for the function of the iron metabolism/ferroptosis regulators examined in our present study (Figure 4) [165]. Many of these proteins are abundant as various proteoforms; this is exemplified by the proteins listed in Appendix A. Future studies, therefore, have to address the limitations of our present study and (i) investigate the complexity of AML also in the uncommon erythroid, megakaryocytic, and myelofibrotic AML variants that were not present among our patients (see Table 1), as well as (ii) additional post-translational modulations and/or the presence of various proteoforms that may differ between leukemic and normal CD34^+^ cell and/or between AML patient subsets. Finally, our study included relatively few patients and our observations therefore need to be verified in larger prospective clinical studies.

Our studies on iron metabolism/ferroptosis show a significantly increased risk of relapse/resistance for a relatively small subset of AML patients having similarities in cellular iron metabolism when compared with normal CD34^+^ bone marrow cells. The patients within this subset are heterogeneous with regard to ELN genetic risk classification [2] but the number of patients in our study is too low to allow for a reliable additional biological characterization of the patients included in this minor subset.

As discussed in Section 4.1, our present observations may be relevant only for AML patients with relatively high levels of AML blasts in bone marrow and peripheral blood.

## 5. Summarizing Conclusions

We compared the proteomic profiles of pretreatment AML (a bone marrow malignancy) cells derived at the first time of diagnosis with a normal counterpart, i.e., normal CD34^+^ bone marrow cells. Based on differentially abundant cellular proteins, we identified three patient subsets that only partially reflected the AML-associated genetic abnormalities. (i) One subset was identified based on the expression of neutrophil differentiation markers and included patients with favorable karyotypes. (ii) Among the other patients without extensive differentiation, we identified a subset with a particularly adverse prognosis and an AML cell profile of iron metabolism/ferroptosis regulators showing similarities with the normal CD34^+^ cells. (iii) The remaining subset showed few signs of differentiation but an altered expression of iron metabolism/ferroptosis regulators.

## Figures and Tables

**Figure 1 proteomes-13-00011-f001:**
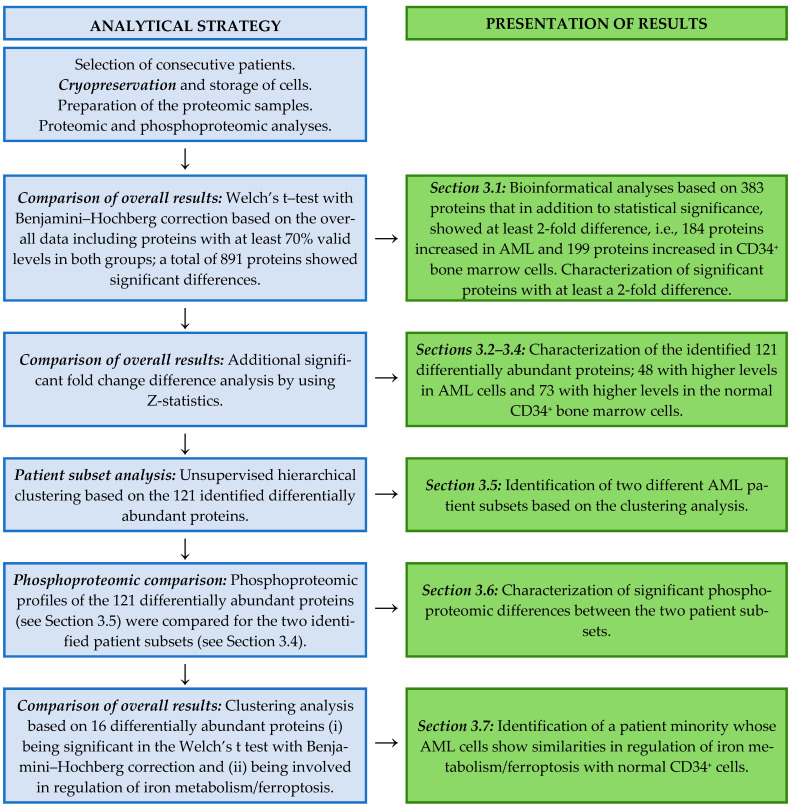
A flow chart showing the various statistical/bioinformatical analyses in this present study and the corresponding presentation of the results in the various subsections of Results.

**Figure 2 proteomes-13-00011-f002:**
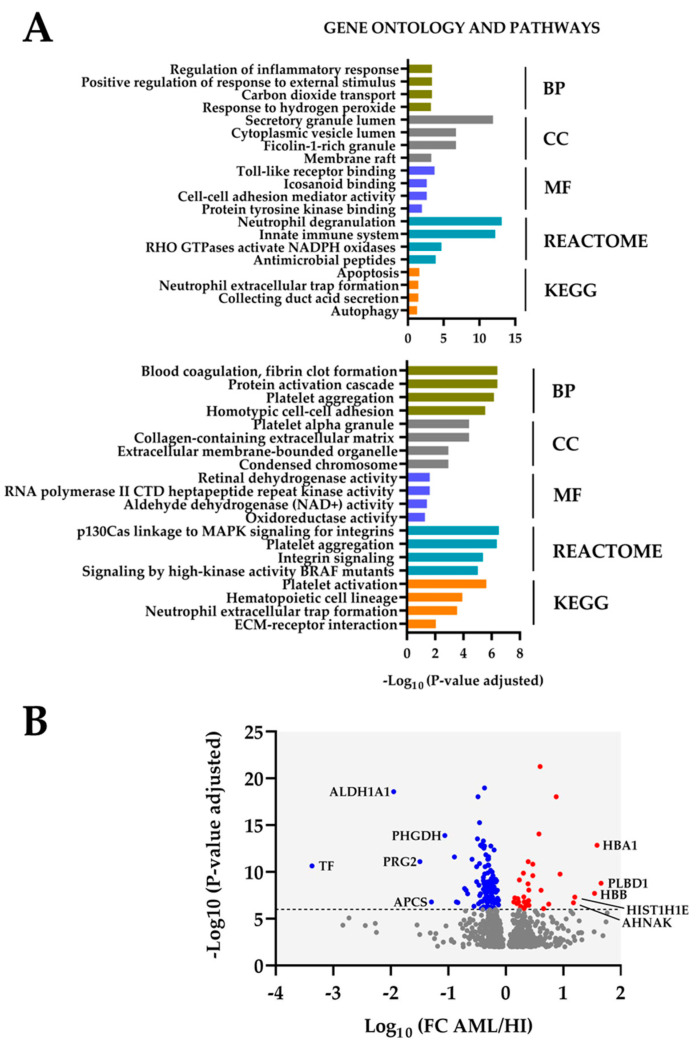
Proteomic differences between primary AML cells derived from 50 patients and normal CD34^+^ bone marrow cells derived from eight healthy individuals. A total of 891 differentially abundant proteins were identified in the initial statistical analysis but only 121 of these proteins showed significant fold change differences (Z-statistics). The analyses presented in the figure are based on these 121 proteins. (**A**) The figure presents the Gene Ontology (GO), Reactome, and KEGG analyses for the 48 (upper plot) and 73 (lower plot) proteins that showed increased and decreased (i.e., increased in normal CD34^+^ cells) levels in primary AML cells, respectively. (**B**) The volcano plot is based on all 891 differentially abundant proteins. The colored dots represent proteins with adjusted *p*-values < 1 × 10^−6^ (see the y-axis); red dots represent increased levels in primary AML cells (i.e., AML cell/CD34^+^ cell > 1.0); blue dots represent proteins with increased levels in the normal CD34^+^ bone marrow cells. Abbreviations: BP, biological process; CC, cellular compartment; FC, fold change; MF, molecular function.

**Figure 3 proteomes-13-00011-f003:**
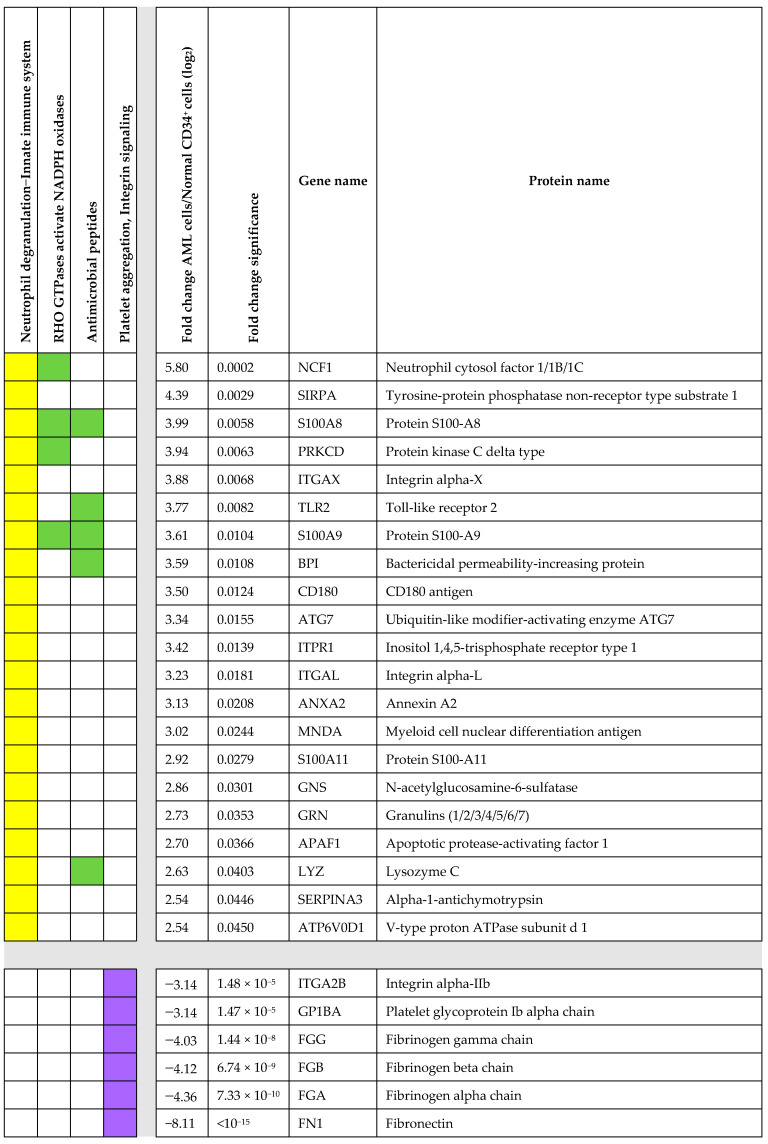
Proteomic comparison of proteins being statistically significant both in the initial Welch’s *t*-test analysis with at least a 2-fold difference and also in the additional AML cell/normal CD34^+^ bone marrow cell fold change comparison (*Z*-statistics, *p* < 0.05)—an overview of proteins included in the top-ranked Reactome terms (see Figure 2A,B). The Reactome terms are given at the top of the figure. The left part of the figure shows the proteins included in the two overlapping terms neutrophil degranulation (R-HSA-6798695, 21 proteins) and innate immune system (R-HAS-168249, 17 overlapping proteins), presented in yellow as a single column, and the two terms (left middle) RHO GTPases activate NADPH oxidase (middle–left, R-HAS-5668599) and antimicrobial peptides (middle–right R-HAS-6803157) presented in green. All these terms were enriched in the primary AML cells. The right column in purple shows all proteins with increased levels in normal CD34^+^ cells and included in the overlapping Reactome terms p130Cas linkage to MAPK signaling for integrins R-HAS-372708, platelet aggregation (plug formation) R-HAS-76009, integrin signaling R-HAS-354192, and signaling by high-kinase activity BRAF mutants R-HAS-6802948 (see Figure 2A).

**Figure 4 proteomes-13-00011-f004:**
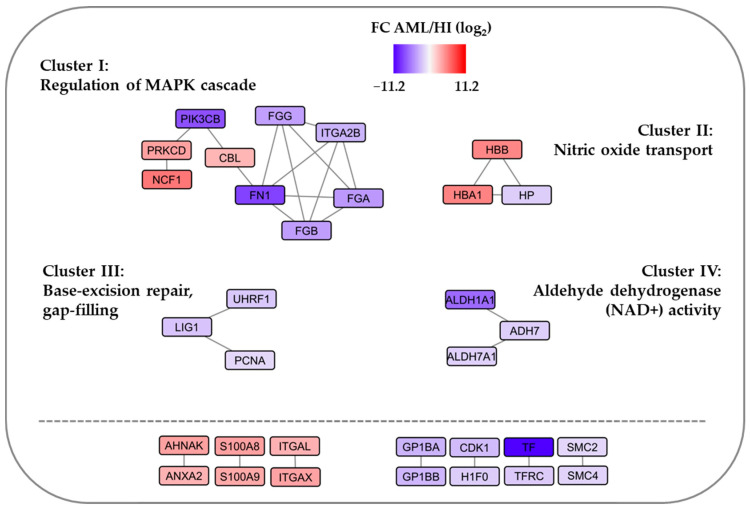
Significant protein–protein interaction networks (**upper part**) and pairs (**lower part**) identified based on the analysis of 121 regulated proteins showing fold change (FC) significance between primary AML cells derived from 50 patients and normal CD34^+^ bone marrow cells derived from eight healthy individuals.

**Figure 5 proteomes-13-00011-f005:**
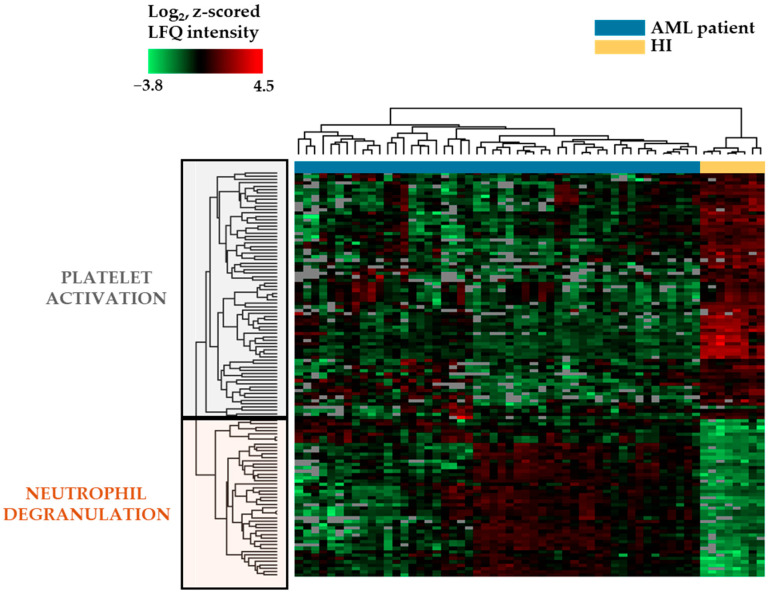
Hierarchical clustering analysis based on the proteomic profile for primary AML cells derived from 50 patients and eight CD34^+^ bone marrow cell populations derived from healthy individuals (HI). The clustering of normal and leukemic cells is shown at the top of the figure; the clustering of the 121 differentially abundant proteins after analysis of fold change significance (*Z*-statistics) is shown in the left part of the figure. The ranking of patients from left to right is presented in Appendix A and the ranking of the proteins from the top to bottom is presented in Appendix A. The two main protein clusters are referred to as platelet activation and neutrophil degranulation due to their inclusion of proteins classified in Reactome terms that reflect these processes (see Appendix A).

**Figure 6 proteomes-13-00011-f006:**
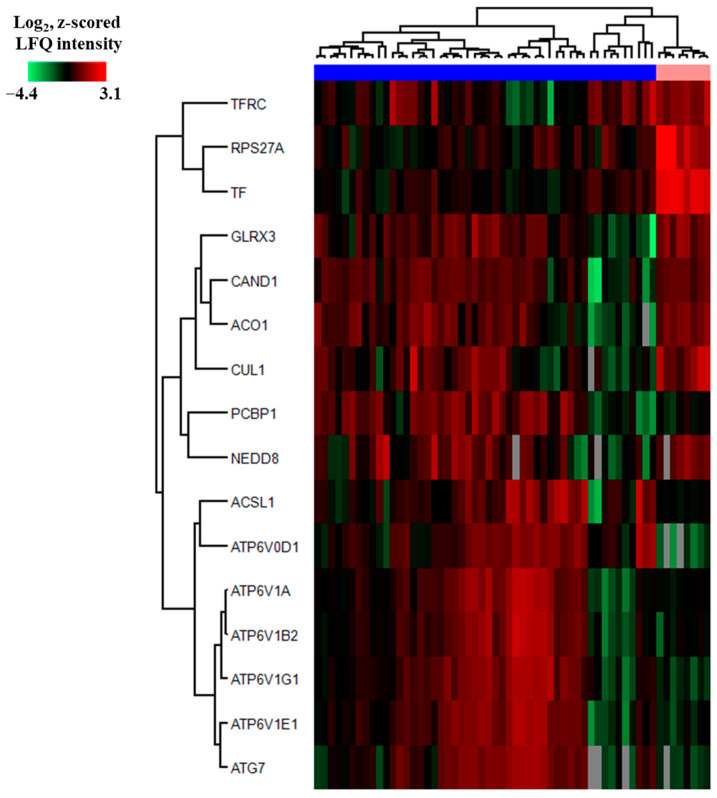
Hierarchical clustering analysis based on the 16 metabolic/ferroptotic proteomic profiles for primary AML cells derived from 50 patients and eight CD34^+^ bone marrow cell populations derived from healthy individuals (HI). This analysis is based on the expression of 16 proteins (i) that were abundant in at least 70% of the AML cell populations or normal CD34+ bone marrow populations, and (ii) differed significantly in the Welch’s *t*-test with Benjamini–Hochberg correction when comparing leukemic and normal cells. The clustering of normal (corn silk color) and leukemic cells (blue color) is shown at the top of the figure. The ranking of patients from left to right is shown at the top of the figure and is also presented in Appendix A. All proteins included in the analysis are known regulators of iron metabolism/ferroptosis according to the Reactome/KEGG classifications. They included (i) three proteins showing significance after the Z-statistics analysis (TF, TFRC, ATP6V0D1); (ii) five proteins showing significance in the Welch’s *t*-test with Benjamini–Hochberg correction and, in addition, a fold change corresponding to >2.0 (ACO1, ACSL1, ATP6V1G1, ATP6V1E1, ATG7); and (iii) eight proteins showing significance in the Welch’s *t*-test with Benjamini–Hochberg correction but having a lower fold change (RPS27A, GLRX3, CAND1, CUL1, PCBP1, NEDD8, ATP6V1A, ATP6V1B2).

**Table 1 proteomes-13-00011-t001:** A summary of important clinical and biological characteristics of the 50 AML patients included in this present study.

Age (median, range, and IQR)	55 years (18–80)		Cytogenetics (number)	Favorable 6
	IQR: 60 − 41 = 19			Intermediate 7
				Adverse 8
Sex (male/female, number)	27/23			Normal 26
				Unknown 3
Secondary AML (number)	Chemotherapy 3			
	MDS 2		FLT3-ITD (number) ^2^	12
			*FLT3*-TKD mutation (number) ^2^	4
FAB classification (number)	M0 6		*NPM1* insertion (number)	15
	M1 9			
	M2 8		ELN 2022 classification	Favorable 14
	M4 15			Intermediate 5
	M5 12			Adverse 13
				Intermediate/adverse 12 ^3^
CD34 positivity (number) ^1^	29			Unclassified 6 ^3^

^1^ CD34 positivity defined as ≥20% positive cells by flow cytometry; four patients not examined. ^2^ Three patients not tested for *FLT3-ITD*, five not tested for *FLT3-TKD* mutation, and three not tested for *NPM1* insertions (TKD, tyrosine kinase domain). ^3^ Extended mutational analyses were available only for 26 of the 50 patients. The group intermediate/adverse prognosis means that these patients did not fulfill any of the ELN criteria for favorable prognosis but they were not examined in an extended mutational analysis with regard to additional adverse mutation as defined by ELN [2]. Unclassified means that the patients were examined neither for all ELN-defined favorable prognostic factors nor for additional adverse prognostic mutations in an extended analysis (i.e., missing karyotype or *NPM1/FLT3/CEBPA* analyses).

**Table 2 proteomes-13-00011-t002:** Protein phosphorylation of primary human AML cells derived from two patient subsets (see Figure 5); the identification of protein phosphorylation sites that (i) were detected in a subset of the 121 differentially abundant proteins (i.e., 38 proteins) that showed at least a 2-fold difference when comparing the levels in AML cells and normal CD34^+^ cells; (ii) in addition, a statistically significant difference when each identified phosphosite was compared between the two patient subsets identified in the clustering analysis is presented in Figure 5 (i.e., 23/28 right versus 18/22 left patients). The first criterion (i) identified 174 phosphosites in 38 of the 121 differentially abundant proteins; 53 of these phosphosites in 16 of the differentially abundant proteins also fulfilled the second criterion (ii) and are listed in the table. The table presents the gene name, protein name, number of phosphosites, and the possible functional importance of the protein phosphorylation status for each of these 16 proteins. The gene names for proteins showing increased levels in normal CD34^+^ cells are underlined and marked in *italics* (see the left column).

Gene Name	Protein Name	Number of Sites	Effects of Protein Phosphorylation
AHNAK	Neuroblast differentiation-associated protein AHNAK	25	Altered compartmentalization and molecular interactions; modulates stem cell differentiation [47,48,49,50]
ANXA2	Annexin A2; Putative annexin A2-like protein	1	Effect on chemosensitivity, possibly also AML prognosis [51,52,53]
APOBR	Apolipoprotein B receptor	3	-
* DBN1 *	Drebrin	1	Actin organization, Mg^2+^ transport [54,55,56]
* DUT *	Deoxyuridine 5′-triphosphate nucleotidohydrolase, mitochondrial	1	Possibly no effect on the enzymatic activity [57]
* GP1BB *	Platelet glycoprotein Ib beta chain	1	-
ITPR1	Inositol 1,4,5-trisphosphate receptor type 1	1	Cellular calcium homeostasis [58,59]
KCTD12	BTB/POZ domain-containing protein KCTD12	1	Cell cycle regulation; cancer support [60]
* MSL1 *	Male-specific lethal 1 homolog	1	Epigenetic regulation in cancer [61]
PBXIP1	Pre-B-cell leukemia transcription factor-interacting protein 1	1	-
PLEC	Plectin	7	Mitosis/centrosome localization [62]
PRKCD	Protein kinase C delta type	2	Phosphorylation profile is important for molecular interactions [63,64,65]
* RCOR3 *	REST corepressor 3	3	-
* SMC4 *	Structural maintenance of chromosomes protein 4	1	Multisite mitosis-associated phosphorylation [66,67,68,69]
* STMN1 *	Stathmin	2	Cancer chemoresistance and progression, stemness marker, and PI3K target. Possibly an adverse prognostic impact in hematological malignancies, including AML [70,71,72,73,74]
* TOP2A *	DNA topoisomerase 2-alpha	2	Regulation of S phase entry, targeted by etoposide and mitoxantrone [75,76]

## Data Availability

All proteomic raw data and MaxQuant output files together with the phosphoproteomic raw data can be found in the ProteomeXchange consortium with the dataset identifiers PXD014997 and PXD058846.

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
