# Peer review of "Proteomic Comparison of Acute Myeloid Leukemia Cells and Normal CD34+ Bone Marrow Cells: Studies of Leukemia Cell Differentiation and Regulation of Iron Metabolism/Ferroptosis"

_proteomes, 2025, doi:10.3390/proteomes13010011_

Round 1
Reviewer 1 Report
Comments and Suggestions for Authors
The authors have presented an interesting, comprehensive study that is of great interest to researchers. I have no comments on the substance of the work, but I have some suggestions on how the results are presented.
1) 3.1. Identification of differentially abundant proteins in AML cells and normal CD34+ cells; an initial analysis based on proteins showing at least a median 2-fold difference. I recommend reformulating the titles of the results sections. Present them more concisely and specifically
2) Figure 2(page 7). What does page 7 mean? Maybe it makes sense to rename it to a table?
3) It makes sense to present panel Fig. 1C as a separate figure, since the reference to it goes far in the text
4) The discussion needs to be structured and divided into subsections. Perhaps the discussion should be shortened (optional).
5) I would recommend that the authors provide a graphical abstract of the study.
Author Response
A complete cover letter is also attached.
The authors have presented an interesting, comprehensive study that is of great interest to researchers. I have no comments on the substance of the work, but I have some suggestions on how the results are presented.
Response: We are grateful for this general comment.
1.1. Section 3.1. Identification of differentially abundant proteins in AML cells and normal CD34+ cells; an initial analysis based on proteins showing at least a median 2-fold difference. I recommend reformulating the titles of the results sections. Present them more concisely and specifically
Response: All titles in the Results section have been rewritten/reformulated.
1.2. Figure 2 (page 7). What does page 7 mean? Maybe it makes sense to rename it to a table?
Response: This comment refers to the legend of the present Figure 3 (now page 10), the figure is presented on the opposite page 9 in the revised version. The intention with this parenthesis is now explained in the same parenthesis.
1.3. It makes sense to present panel Fig. 1C as a separate figure, since the reference to it goes far in the text
Response: We are grateful for this suggestion. We have created a new and separate Figure 4 from the original Figure 1C. Later figures have been renumbered.
1.4. The discussion needs to be structured and divided into subsections. Perhaps the discussion should be shortened (optional).
Response: We have restructured the Discussion section. The template for Proteasomes does not give details with regard to the structure of the Discussion section, and for this reason we suppose that division into separate sections is acceptable. The Discussion section now has four subsections; i.e. methodological consideration, AML cell differentiation, iron metabolism/ferroptosis and limitations of the study. We hope this solution can be accepted.
The overall length of the original parts was 2051 words, the present length with new and longer headings is 2100. Thus, there is an overall shortening of only 50 words. However, based on recommendation from the editor of this Special Issue we did several modifications of the original Discussion section before our manuscript was sent out for review. For this reason we have been careful with extensive revision of the original text in the Discussion, especially the last parts.
Based on the comments from the other reviewers we have also added new parts to the Discussion; these are marked with yellow.
1.5. I would recommend that the authors provide a graphical abstract of the study.
Response: This has been done and is included in the new submission. A draft for a graphical abstract is enclosed to this letter.

Reviewer 2 Report
Comments and Suggestions for Authors
This manuscript describes the proteomic comparison of AML cells and normal CD34+bone marrow cells with published data. Although the sample size is limited, the data from clinical samples can always give some insights for the understanding of the mechanism of AML. However, I have the following concerns/comments:
1. Can not find dataset corresponding to PXD058846
2. Which paper the CD34+ normal bone marrow cell data was from? PXD056015 cannot give any result from proteome Xchange.
3. Section 2.2: Basically, the sample preparation and LC-MS/MS were described here, but not in a clear way. Since this is the re-analysis of already published data, the emphasis should be put on how the mass spec data was analyzed instead of how the mass spec data were generated. How the data was searched, normalized and quantified were missing.
4. In Table S5: why the Raio (AML/CD34+) can be negative? Are they log ratio? Can you explain why this ratio value is different from the value in Table S2 for the same protein?
5. In Fig 1, there are both log10FC and log2FC, this should be consistent
6. In Fig 2: why FC is negative?
Author Response
A complete cever letter is also attached
This manuscript describes the proteomic comparison of AML cells and normal CD34+bone marrow cells with published data. Although the sample size is limited, the data from clinical samples can always give some insights for the understanding of the mechanism of AML. However, I have the following concerns/comments:
Response: We are grateful for the general comment.
2.1. Can not find dataset corresponding to PXD058846
Response: We have clarified this. Please log in to https://www.ebi.ac.uk/pride/login at the PRIDE website, the reviewers can the access the dataset by using the following account details:
Username: PXD058846
Password: uVgsDD31dv4O
2.2. Which paper the CD34+ normal bone marrow cell data was from? PXD056015 cannot give any result from proteome Xchange.
Response: The proteomic profiles of the CD34+ bone marrow cells have not been published in any previous article. They are analyzed in this manuscript for the first time. Please ignore PXD056015. The project PXD058846 contains raw data for cryopreserved normal CD34+ bone marrow cells from eight healthy Caucasians (AML-OSvsNS-LF_ctr1-8.raw). The submitted samples have been analysed together with raw files derived from 50 different AML patient samples (9 new patient samples in this project PXD058846 and 41 patient samples submitted to project PXD014997 in 2020. We have substituted PXD056015 by PXD058846 in the revised manuscript (page 3 and 4).
2.3 Section 2.2: Basically, the sample preparation and LC-MS/MS were described here, but not in a clear way. Since this is the re-analysis of already published data, the emphasis should be put on how the mass spec data was analyzed instead of how the mass spec data were generated. How the data was searched, normalized and quantified were missing.
Response: We thank you for the chance to add more information on this point. We have added more details on the analysis of LC-MS/MS data on page 4 and in the form of a flow chart in Figure 1 (page 5).
2.4. In Table S5: why the Ratio (AML/CD34+) can be negative? Are they log ratio? Can you explain why this ratio value is different from the value in Table S2 for the same protein?
Response: We agree that the presentation of the data was very complicated in our original article and needed to be clarified and simplified. This was also commented by reviewer 3.1 that wanted us to prepare a flow chart for our data analysis. The following alterations have been made:
- We have prepared a flow chart presented in Figure 1 (page 5)and giving an overview of our statistical/bioinformatical analyses.
- Tables S1 and S2 now present the proteins also identified in the Z-statistics analysis presented in Table S5 (yellow color on the gene name). The fold change in this table represents the absolute mean values (without log transformation), whereas Table S5 presents the median fold change (FC) as log 2 values. The statistical strength of each identified protein is still documented by p- and q values. We clearly state that we have used a fold change cut-off of 2. This means a fold change exceeding 2, i.e., 2 in favor of the AML cells for Table S1/S3 and 2 in favor of the CD34+ cell (AML/CD34+ ratio <0.5) in Table S2/S4.
- Fold change data are relevant for example for the protein interaction network analyses, and they are therefore presented also in Table S5 that lists the proteins that were significant also after Z-statistics analyses.
- It is now indicated in Tables S1 and S2 by yellow color which proteins that remained significant after the analysis by Z-statistics.
- It is clearly stated (Table S5, Figure 2, Figure 3) when we use log values and what kind of log values.
- The selection of proteins for the iron metabolism/ferroptosis studies is now described more in detail in the legend to Figure 6 (page 15).
- The editor of this special issue evaluated the manuscript before the review process, and he wanted us to give this detailed description of the proteomic methodology. For this reason we have not altered these parts.
2.5 In Fig 1, there are both log10FC and log2FC, this should be consistent
Response: This refers to the present Figure 2. We are grateful for this comment so that we can clarify the use of log2 or log10 of the fold change (FC) in volcano plots. We think both log2 and log10 are equally used in volcano plots of proteomics data. The reason why we have used log10 fold change is for not having very disperse points in the plot as the fold change vary quite a lot in the dataset. We did volcano plots using log10 and log2 FC and we found that the log10 FC plot was better. As long as logs are well indicated, log2 or log10 of the fold change in the different plots, as we have done, this should in our opinion not make any confusion. As stated above we think both are equally used in volcano plots of proteomics data.
Only log10 is now used in the present throughout the new Figure 2. In the new Figure 4 we use log2, but we hope this can be accepted.
2.6 In Fig 2: why FC is negative?
Response: In these cases the fold change values are presented as log2 LFQ intensities of the AML and CD34+ cells. We have now clarified this in the column head; fold change (FC) AML cells/Normal CD34+ (log2). As the ratio is of log2 values, positive and negative values are possible.

Reviewer 3 Report
Comments and Suggestions for Authors
The paper is well written and analyses many aspects of acute myeloid leukemia in depth. The aim of the study, which is well described in the introduction, is also supported by relevant context on the importance of cell differentiation, iron metabolism and necroptosis. The description of the methods, such as the use of mass spectrometry and statistical analyses, is detailed and well documented.
I see this paper as a preliminary investigation, and I would then enlarge the cohort. Just some minor comments:
- In order to better explain the patients clusterization workflow, in my opinion, a flowchart could be useful
- In my opinion, as reported, the paper represents a preliminary investigation, and I would add this limit also in “introduction”.
- In my opinion, the fact that new therapies allow treatment of frailty patients (e.g., elderly patients) is an important point. The cited articles are well incorporated in the paper, are, however, articles from a few years ago, I would suggest citing these important works published in the 2024 (PMID:39312920; 38343151).
- Line 58: typo error, “promyelocytib” instead “promyelocytic”.
- We report that the patients included in this analysis are “from the same geographical area during a defined time period and receiving intensive conventional AML therapy”. The questions are (i) how long is period?, (ii) in this cohort are there patients underwent allogenic or autologous stem cells transplantation?
- We report “with a high percentage of AML blasts among circulating leukocytes”. Did or did not consider bone marrow aspirate? If no, please explain.
- Line 91: please report the protocol used.
- Line 97: the authors report median age. I suppose that is not normally distributed. Please do not report range but IQR, for major statistical rigor. Also in the table #1.
- Please report in table #1 the risk stratification as per ELN 2022.
- Also in table #1 the authors report standard molecular biology (FLT3-ITD and NPM1). Why for NPM1, only the insertion was reported? Regarding FLT3 mutational status, why TKD not reported?
- Also in table #1, why four patients not evaluated for CD34? And why were three patients not tested for FLT3-ITD and NPM1.
- Line 151: please report GraphPad version.
- Why 2016 WHO classification was used? My suggestion is to use WHO 2022. In addition, the morphological classification that was used is the FAB classification.
- Lines 435-436: please report IQR and not range.
I hope my suggestions will help the authors in improving the manuscript.
Author Response
Please see the attached complete cover letter.
The paper is well written and analyses many aspects of acute myeloid leukemia in depth. The aim of the study, which is well described in the introduction, is also supported by relevant context on the importance of cell differentiation, iron metabolism and necroptosis. The description of the methods, such as the use of mass spectrometry and statistical analyses, is detailed and well documented.
Response: We are grateful for this general comment.
3.1. In order to better explain the patients clusterization workflow, in my opinion, a flowchart could be useful
Response: A flow chart is now presented in the new Figure 1 (page 5).
3.2. In my opinion, as reported, the paper represents a preliminary investigation, and I would add this limit also in “introduction”.
Response: This is now added in the Introduction (page 2, last chapter of the Introduction). We have also stated this in a short additional comment in Section 4.4 (page 21).
3.3. In my opinion, the fact that new therapies allow treatment of frailty patients (e.g., elderly patients) is an important point. The cited articles are well incorporated in the paper, are, however, articles from a few years ago, I would suggest citing these important works published in the 2024 (PMID:39312920; 38343151).
Response: These two articles are now included as references 5 and 6 (see page 2). We left out the original reference 5 that was a review article from 2016, but we kept one of the original reference (now reference 4) that is an updated review from 2023. We hope these solutions can be accepted.
3.4. Line 58: typo error, “promyelocytib” instead “promyelocytic”.
Response: This has been corrected (page 2, second chapter).
3.5. We report that the patients included in this analysis are “from the same geographical area during a defined time period and receiving intensive conventional AML therapy”. The questions are (i) how long is period?, (ii) in this cohort are there patients underwent allogenic or autologous stem cells transplantation?
Response: The time period is now given (Section 2.1 page 2).
None of the patients received autologous stem cell transplantation, this is stated in Section 2.1. (page 2), together with a new statement referring to information about allogeneic stem cell transplantation in Table S8 and S13.
3.6. We report “with a high percentage of AML blasts among circulating leukocytes”. Did or did not consider bone marrow aspirate? If no, please explain.
Response: To further elucidate the patient characteristics with regard to bone marrow and circulating AML cells we have now included analyses of percent AML blasts among nucleated bone marrow cells and levels of circulating blasts (i.e. degree of leukemization) in our analyses/comparisons of AML patient subsets identified in the two clustering analyses (Figures 5 and 6). The results are described in Sections 3.5 (page 12) and 3.7 (page 16) and they are commented in Sections 4.1 (pages 17-18) and end of 4.2 (page 20).
3.7. Line 91: please report the protocol used.
Response: The protocol for freezing and thawing is now briefly described in Section 2.1 (page 3, chapter 2). We also discuss our methodological strategies, including the use of cryopreserved cell, in a separate part of the new Section 4.1 (pages 17-18).
3.8. Line 97: the authors report median age. I suppose that is not normally distributed. Please do not report range but IQR, for major statistical rigor. Also in the table #1.
Response: We agree in this comment with regard to the statistic. However, we think that the range also has a value (although limited) as easily available information/documentation of the fact that we only included adult patients and a few rather old patients. It may also have an interest to see where in the main clusters the oldest patients located.
For this reason we report both the Interquartile range and the variation range together with median values throughout the article, including those parts pointed out by the reviewer including Tables 1 and S9). We hope this solution can be accepted (see also comment 3.13).
3.9. Please report in table #1 the risk stratification as per ELN 2022.
Response: This information is now reported in Table 1 as suggested by the reviewer. Extended mutational analyses were not available for several of the earlier patients and biological material for additional genetic analyses is no longer available. Several patients therefore had to be classified as intermediate/adverse or unclassified; this is explained in (i) a new footnote 3 in Table 1, (ii) an additional comment in Material and methods (Section 2.1, pages 2-3 ). The ELN classification of the 10 exceptional patients identified in the ferroptosis/iron metabolism clustering in Figure 6 is also commented in Section 3.7 (page 16).
We hope the solution presented in Table 1 with regard to the ELN classification can be accepted.
3.10. Also in table #1 the authors report standard molecular biology (FLT3-ITD and NPM1). Why for NPM1, only the insertion was reported? Regarding FLT3 mutational status, why TKD not reported?
Response: We agree in the NPM1 comment, but unfortunately this is the only NPM1 analysis that is available for our patients and we do not have available material for additional genetic analyses.
Our results for the data for the FLT3 tyrosine kinase domain mutations are now included in Table 1; these results are also reported in Tables S8 and S13.
3.11. Also in table #1, why four patients not evaluated for CD34? And why were three patients not tested for FLT3-ITD and NPM1.
Response: All our results for CD34 positivity are from flow cytometric analyses of freshly isolated cells, and CD34 was for a period left out from our routine panel for certain patients because it was regarded to have no prognostic impact. As pointed out in our methodological discussion Section 4.1. cryopreservation will reduce the level of several cell surface molecules (see our reference Sasnoor, J Hematother Stem Cell Res 2003;12:553-64; PMID: 14594512.). For this reason additional analyses based on cryopreserved cells were not done because in our opinion this may not be comparable with the analyses for the rest of the patients. Thus, in our opinion the best solution is to regard this as “not tested”.
For those few patients not tested for FLT3 and NPM1 genetic abnormalities additional material for these analyses is no longer available. Our priority for available material to this study was the proteomic and the phosphoproteomic studies; especially the last ones required relatively large amount of biological material.
We hope these explanations can be accepted, and that the lack of these data (only for relatively few patients) will not be decisive with regard to the final concluding evaluation of our article.
3.11. Line 151: please report GraphPad version.
Response: We have added this information (Section 2.4, page 6).
3.12. Why 2016 WHO classification was used? My suggestion is to use WHO 2022. In addition, the morphological classification that was used is the FAB classification.
Response: We refer to the 2016 WHO classification as it appears in the WHO textbook. Both the WHO 2016 and 2022 have a class named “AML not otherwise specified”. In the WHO 2016 textbook this class follows the FAB classification, i.e. detailed morphological/biological/diagnostic criteria are described and it is also stated how the various subsets of each classes correspond to the FAB classification. For this reason we prefer to refer to this updated detailed presentation in WHO 2016 rather than the recent WHO 2022 classification and the original FAB classification.
We have now altered our original reference 95; instead of referring the WHO 2016 article in Blood we now refer to the article in the WHO book that includes this detailed WHO-2016/FAB presentation.
Our referring to WHO 2016 is also explained in the first chapter of Section 4.1 (page 17, first chapter of the section). We hope this explanation and our solution can be accepted.
We would finally emphasize that the new WHO 2022 classification is also referred to in our article (reference 1).
3.13. Lines 435-436: please report IQR and not range.
Response: We agree in this comment with regard to the statistic. However, as commented above we think that the range also has a value (although limited) as easily available information/documentation of the fact that we only included adult patients and a few very old patients. It may also have an interest to see where in the main clusters the oldest patients located.
For this reason we report both the Interquartile range and the variation range together throughout the article. We hope this solution can be accepted (see also comment 3.8).

Round 2
Reviewer 2 Report
Comments and Suggestions for Authors
The authors addressed all my questions. Current Fig 1 can be improved.
Comments on the Quality of English LanguageEnglish can be improved.
Author Response
Comment: The authors addressed all my questions. Current Fig 1 can be improved.
Response: We agree in this comment. Because the two other reviewers did not have any comment, the content of the figure is the same but it has been redesigned. The figure has two columns; this is now indicated by different colors and each column now has a heading. For each box we indicate the key words with bold and italics.